# Tillage Methods Change Nitrogen Distribution and Enzyme Activities in Maize Rhizosphere and Non-Rhizosphere Chernozem in Jilin Province of China

Ning Huang [1,2], Xingmin Zhao [1,2,*], Xinxin Guo [3] , Biao Sui [1,2], Jinhua Liu [1,2], Hongbin Wang [1,2] and Jialin Li [1,2]

1   College of Resource and Environment, Jilin Agricultural University, Changchun 130118, China;
    huangning@jlau.edu.cn (N.H.); suibiao@jlau.edu.cn (B.S.); liujinhua80@126.com (J.L.);
    asionwang@163.com (H.W.); lijialin@jlau.edu.cn (J.L.)
2   State Key Laboratory of Improvement and Utilization of Saline-Alkali Soils (Inland Saline-alkali Land of
    Northeast China), Ministry of Agriculture and Rural Affairs of China, Changchun 130118, China
3   Department of Environmental Engineering, Faculty of Engineering and Green Technology, Universiti Tunku
    Abdul Rahman, Kampar 31900, Malaysia; guox@utar.edu.my
*   Correspondence: zhaoxingmin2022@163.com

**Abstract:** The tillage method in farming systems is essential to develop strategies to increase fertilizer uptake by plant roots and to avoid environmental pollution. The field study aimed to investigate the characteristics of nitrogen and enzyme activities in rhizosphere soil with different tillage methods. Four treatment plots applied with fertilizers were established: continuous rotary tillage (CR), plowing-rotary tillage (PR), continuous no-till (CN) and ploughing-no-till (PN). The total content of nitrogen in chernozem was high during early stages of plant growth, and then it decreased with the maize growth. In the rhizosphere soil, the total N accounted 1314.45, 1265.96, 1120.47, 1120.47, 1204.05 $mg \cdot kg^{-1}$ of CR, PR, CN, and PN, respectively, which were markedly greater than that of non-rhizosphere soil (1237.52, 1168.40, 984.51, 1106.49 $mg \cdot kg^{-1}$ of CR, PR, CN, and PN, respectively). At first growth stages, content of $NH_4^+$-N and $NO_3^-$-N in two soil regions was low, then increased gradually, which followed the order of CR < PR < PN < CN. The rhizosphere soil showed slightly higher concentration of $NH_4^+$-N and $NO_3^-$-N than non-rhizosphere. The soil enzymes were more active in the rhizosphere soil than that of non-rhizosphere during the whole maize growth stages. Due to minimal damage to the soil environment and optimal soil moisture and temperature, the urease and catalase activities were greatest in the rhizosphere for CN treatment. Therefore, CN was recommended to be used by farmers for the improvement of macronutrient availability and soil enzyme activities in the soil.

**Keywords:** rhizosphere soil; non-rhizosphere soil; tillage methods; nitrogen; enzyme activities

## 1. Introduction

In relation to crop growth, soil can be differentiated into two kinds, which are rhizosphere soil and non-rhizosphere soil [1]. Defined, "rhizosphere" is the thin area of soil surrounding plant roots. The rhizosphere can also be explained as a combination of solid particles around the root zone and a variety of changing community of soil microbes, specifically bacteria [2]. The non-rhizosphere region, sometimes called bulky soil, is the soil layer free of plant roots and microbes and not part of any rhizosphere region [3]. Many microbial processes take place in rhizosphere region because the secretions from roots act as principal food sources for microbes [4]. Van der Voortet al. [5] found that in rhizosphere area, microorganisms provide a variety of ecosystem beneficial processes to plant, such as nutrient gaining, soil sustainability, and resistance to alarms. Root excretions of plants secrete about 10–20% of their photosynthate, which improve the growth and physiological activities of different bacterial and fungal groups in the rhizosphere region [6]. Consequently, these helpful microorganisms are able to promote plant growth by increasing

nutrient uptake, generating auxins, and protecting them from physical stresses [7]. Rhizo-sphere processes have been extensively researched, but little concentration has been paid to process-based rhizosphere management at agro-ecosystem level [8]. The soil chemical activities taking place in the rhizosphere region are determinants of the movement and adsorption of soil nutrients with microbial variations and also control the ability of the plant to utilize the available nitrogen, and hence significantly improve farming system productivity and sustainability [9,10]. Deng and Tabatabai [11] reported that soil bulky density have negative relationship with soil enzyme activity.

Land productivity and soil fertility depend on factors like soil quality, fertilizers, management practices, and the type of crops. The application of nitrogen, phosphorus, and potassium increase the natural supply of nutrients for plants and generate higher yields, and nitrogen plays a very important role among them [12,13]. In China, continuous crop production is a threat to the soil environment, like reduced organic matter, soil carbon, soil microorganism, and soil fertility depletion in large [14]. Farmers in China and elsewhere desire higher yields. Therefore, over-fertilization has become a common practice. However, too much fertilizer does not always lead to high yield [15]. Instead, it causes environmental problems and other economic problems [16]. Zhang et al. [17] reported that from 1998 to 2009, the crop production raised up to 10%, while the application of chemical N fertilizer increased by 49%. This implies that in the past ten years, the acceleration in fertilizer use did not result in the corresponding production. Recently, dealing with agricultural nutrients inputs such as fertilizer to provide optimum yield and protect the environment is one of significant problems in agriculture.

Soil enzymes serve an essential function in farming systems as they work for various necessary biochemical reactions and life-givers of microbes with the conservation of soil structure, and the breakdown and development of nutrient mineralization and organic matter. Therefore, soil enzymes are tools that can protect soil ecosystem. Numerous scenarios that influence soil-plant-microorganism and their interaction, in turn, affect soil enzymes' work and land productivity [18,19]. Burns [20] reported that soil enzymes are originated from microbial, being derivative of intracellular, cell-associated, or free enzymes. Almost all soil biochemical processes are controlled by soil enzymes and are sensitive reaction to fluctuations in tillage management practices [21]. It has been observed by Hamido and Kpomblekou-A [22] that phosphatase and urease activities in the top layer of soil substantially improves under narrow tillage, because available nutrient for enzyme metabolism was the maximum. Mu et al. [23] reported that the exhaustion of soil densification might have slightly substantial effects on significant soil enzyme activity via the enhanced root density. They are catalysts with several vital reactions in the soil, which were useful for microorganism activities and soil structure equilibrium, organic materials breakdown, humus formation, and nutrient cycling, and therefore playing an essential role in sustaining soil ecology, physicochemical properties, and fertility [24,25]. It is crucial to explore nutrients, bacterial richness, enzyme activity, and microbial community in rhizosphere [26].

The effective tillage methods are fundamentally assessed in two groups: traditional tillage methods and conservational tillage methods. Conservational tillage involves systems like reduced tillage, no tillage, mulch till, ridge-till, and line tillage [27]. No-till planting or minimum tillage can conserve the soil, reduce erosions, and decrease fuel consumption [28]. Iqbal et al. [29] found that zero tillage significantly enhances many soil characteristics, whereby extreme and unessential tillage activities give acceleration to inverse phenomena that are negative effect on the earth. Also known as direct seeding, no-till, and sometimes conservation tillage/conservation agriculture are becoming the dominant practice in modern agriculture, effectively protecting soil from erosion while providing the best economic returns and enhanced environmental benefits. Hence, there is a vital interest and importance in the move from excessive tillage to conservation and zero tillage approaches with the purpose of managing the erosion process. Rashidi and Ke-

shavarzpour [30] demonstrated that conventional tillage practices might result in evolution in soil structure by improving moisture content and bulk density.

Constant disturbance by traditional tillage cause finer and loose-setting soil structure, while conservation and no-tillage prompt soil hard. Long-term narrow tillage created a hard-plowing pan and facilitated subsoil compaction, which prohibits root penetration and decreases nutrient and water uptake from inner layers, therefore resulting to overcoming the effect of drought and yield improvement [31]. It has been observed by many studies that rotation of tillage approaches has been recognized as the best way of resolving problems caused by continuous monoculture [32,33]. Many studies have revealed that vertical gradient of soil properties reduced under deep tillage due to improve of soil nutrient levels in 20–60 cm layer of soil by put in the ground most organic materials [23,34]. The deep tillage method causes increased carbon and nitrogen mineralization by destructing macro-aggregates structure of soil and improving soil metabolic strength with enhancement in soil air and profile [30,35]. Adoption of proper agricultural management practices, however, may sustain soil productivity tillage method is among of the most common practices in agricultural production that affects soil properties such as nutrient status which can improve soil productivity. An early research was observed that rotational tillage method could promote the amount of organic material and total nitrogen of soil, recommending that rotation of shallow tillage and deep tillage is good practice for continuous wheat fields cropping [36]. Different tillage practices have effects on soil properties in a calcic haploxeralf in a leguminous–cereal rotation system and demonstrated that the degree and period of effects on soil properties could be continued for 2 years. Tillage and cultivation methods changes soil properties and lead to soil fertility depletion and land productivity. Proper agricultural management practices may sustain soil productivity [37].

Cropland soils in China are varying degrees of acidification due to overuse of nitrogen fertilizers, decreasing productivity and contributing certain amounts of greenhouse gas emissions. Chemical fertilizer overuse is major source of environmental pollution from agriculture. Farming practices can alter the physical and chemistry environment of the soil and inevitably affect the distribution of nitrogen and enzymes in the rhizosphere. Black soil protection and conservation tillage have been widely concerned in China. The expectant aera of conservation tillage account for about 70% of the total area of cultivated land in suitable areas in Northeast China by 2025. However, due to great differences in climate resources and soil types, it is not realistic to completely rely on conservation tillage technology to protect black soil. In recent years, some new farming methods with "straw returning to the field" as the core have been developed. It mainly includes four major farming methods: non-tillage, less tillage, straw breaking and mixed tillage and straw deep turning tillage. However, the technical effects of different farming methods, fertilizer utilization rate and regional adaptability need to be further studied. This work seeks to find distribution of nitrogen and enzymatic activities in rhizosphere and non-rhizosphere soil of maize with diverse tillage. The results will provide a theoretical basis for the establishment of rational straw returning and tillage methods in the study area.

## 2. Materials and Methods

### 2.1. Research Design

The research was organized in Taobei District of Qingshan Town in Baicheng City, Jilin Province, China (N 45°41′, E 122°55′), about 200 m altitude. Soil type was sandy, loamy chernozem. Chemical properties of 0–20 cm soil layer were mentioned in (Table 1) below. The experiment took place over three years (2019–2021), including continuous rotary tillage (CR), continuous no-tillage (CN), plowing-rotary tillage (PR), and plowing-no tillage (PN). CR: rotary tillage every year, the soil tilling depth was approximately 10–12 cm, no-tillage seeder sowing, and fertilizer; CN: seeder sowing and fertilizer at approximately 20 cm soil depth, direct use of a no-tillage seeder for sowing and fertilization without other treatments; PR: plowing at a depth of approximately 20 cm in the first year and the same treatment as CR in the last two years; PN: plowing and tillage at approximately 20 cm

depth in the first year and the same treatment as CN in the second year. There were three replicates per treatment in 12 subplots of 500 m$^2$. The corn stalks are returned to the field in autumn. In May, the maize was planted and harvested in October. Accompanied by seeding, 800 kg·ha$^{-1}$ combined fertilizer (N: P$_2$O$_5$:K$_2$O = 26%:11%:11%) was applied.

**Table 1.** Basic physicochemical properties of the tested soil.

| Characteristics | Value |
| --- | --- |
| Organic matter | 16.97g·kg$^{-1}$ |
| pH | 7.10 |
| Available P | 34.64 mg·kg$^{-1}$ |
| Total P | 462.08 mg·kg$^{-1}$ |
| Total N | 1.52g·kg$^{-1}$ |
| Alkaline N | 65.83 mg·kg$^{-1}$ |
| Available K | 75.99 mg kg$^{-1}$ |
| Total K | 21,606.16 mg·kg$^{-1}$ |
| Soil bulk density | 1.61 g·cm$^{-3}$ |
| Soil texture | Black Sandy loam |
| Type of the soil | Black Chernozem |

### 2.2. Soil Sampling

The chernozem from CR, PR, CN, and PN treatments was sampled at four maize growing stages in 2021 October, namely the seeding stage, elongation stage, tasseling stage, and maturity stage. A soil sample was speared into two components, rhizosphere and non-rhizosphere of soil. After sowing, at each specific time, all the fresh soil samples were brought to the laboratory and dried naturally in the air, and then ground into a fine powder (passed through a 0.83 mm, 0.25 mm, and 0.15 mm mesh screen).

The whole plant root with apical and older root's part were dug out in each prominent sub-plot and each root. The soil was removed carefully and systematically with a soil mass of 28 cm (14 cm on each side of the plant base in intra row direction) × 35 cm (10 cm in narrow inter-row and 25 cm in wide inter-row) and a depth of 40 cm. Following the careful removal of unsecured carried soil (collected as non-rhizosphere soil), the prevailed firmly held earth was shaken gently over an unsoiled paper sheet. After careful hand picking out the visible thin roots (except root hair cells), the soil was collected as rhizospheric soil. The collected soil samples were grounded down into fine particles and sieved through a 3 mm sieve. Appropriate amounts of ground soil were used to determine the total N, NH$_4^+$-N, NO$_3^-$-N, and soil enzyme concentrations.

### 2.3. Test and Analyze Methods

The total nitrogen was determined using the Kjeldahl method [22]. The enzyme activity were determined by the Du method [38].

NH$_4^+$-N and NO$_3^-$-N: 10 g (dry weight equivalent) of fresh soil was mixed with 60 mL of distilled water in plastic bottles, and incubated for 18 h at 70 °C. The samples were subsequently shaken for 5 min in an orbital incubator and centrifuged at 10,000 RPM for 10 min, and the supernatant was filtered through a 0.45 μm Millipore membrane filter (MF-Millipore Merck, Darmstadt, Germany) using a syringe. The concentrations of NH$_4^+$-N and NO$_3^-$-N in the filtrates were measured using a continuous-flow analyzer (Ling Kong, LM chu).

Urease activity: Urease activity was analyzed by 2 h incubation of a reaction mixture of 2.0 g of field-moist soils and 2.5 mL of 80 mmol/L urea solution at 37 °C. Deionized water was added to the controls. The ammonium released was extracted with 50 mL KCl solution and product was measured at 690 nm with a digital UV-Vis spectrophotometer against the reagent blank.

Alkaline phosphatase: The substrate for alkaline phosphatase was 0.05 mmol/L p-nitrophenyl phosphate (p-NPP) which was prepared in pH 11 buffer. Then, 1 mL

0.05 mmol/L p-NPP solution and 4 mL pH 11 buffer were added to 2.0 g soil samples, and the flasks were capped and incubated at 37 °C for 1 h. After incubation, 1 mL 0.5 mmol $CaCl_2$ and 4 mL 0.5 mmol NaOH were added, filtered after shaking. The p-nitro-phenol (PNP) in the filtrate was determined in a spectrophotometer at 400 nm wavelength.

Catalase activity was measured by back-titrating residual $H_2O_2$ with $KMnO_4$. For this, 2.0 g of soil samples were added to 40 mL distilled water with 5 mL of 0.3% hydrogen peroxide solution. The mixture was shaken for 20 min and then 5 mL of 1.5 mol/L $H_2SO_4$ were added. Afterwards the solution was filtered and titrated using 0.02 mol/L $KMnO_4$. The reacted amount of 0.02 mol/L $KMnO_4$, calculated per gram of dry soil, was used to express the activity of catalase.

The data were analyzed, and the differences compared using SPSS Statistics 23.0 (SPSS, Inc., Chicago, IL, USA). The means were compared by Duncan's test at the 0.05 significance level. Figures were presented by Origin Pro 8.0. Means and standard errors from the statistical analysis were brought into Origin Pro 8.0, and figures were then created using the column tool.

## 3. Results

### 3.1. Total N Dynamics in Maize Soil with Various Tillage Methods

After fertilization, the total nitrogen contents were high for all treatments (Figure 1). The total N was a bit different among the four tillage methods, which decreased in the order of CN < PN < PR < CR during all stages of maize growth. For CR and PR treatments, the difference of total nitrogen content was significant only in sowing period, but not significant difference in other periods. Comparison between different tillage methods studied, in both soil regions, CN treatment had the lowest total N content than that of other treatments. During seedling stages, the total N contents were high, then gradually reduced with plants growing until physiological maturity, which accounted for the lowest total N in all the treatments. The application of fertilizer had a vital effect on total N in the rhizosphere and non-rhizosphere soils. Total N in rhizosphere soil was higher than non-rhizosphere soil, indicating that most of the nutrients shifted to the rhizosphere region of the soil under all the treatments.

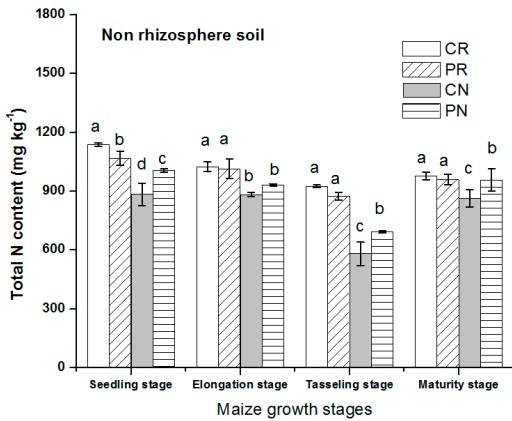 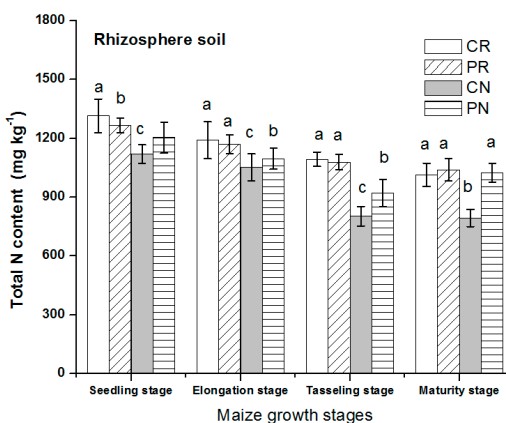

**Figure 1.** Dynamic change of total N in rhizosphere and non-rhizosphere soil during maize growth stage. Notes: lowercase letters "a, b, c, d" indicate significant differences among different treatments, different lowercase letters above the bars indicate significant differences of total N content.

### 3.2. Dynamic Change of $NH_4^+$-N and $NO_3^-$-N in Chernozem under Different Tillage Methods

At early stages, $NO_3^-$-N amount was low, then slightly increased to the highest level at the harvest stage in rhizosphere and non-rhizosphere soil (Figure 2). The seedling growth stage had a relatively lower amount of nitrate activities than maturity stage. In comparison with rhizosphere soil region, the non-rhizosphere soil region had higher $NO_3^-$-N content. This indicated that more $NO_3^-$-N was used up by the plant. The CN treatments had the lowest level of $NO_3^-$-N than the others. At the harvesting period, under the rhizosphere

soil, the CN treatment had 42.4%, 71.5%, and 79.5% greater than PN, PR, and CR treatments, respectively. Similarly, in the non-rhizosphere soil, the CN treatment had 16.7%, 75.4%, and 95% greater than that of PN, PR, and CR treatments.

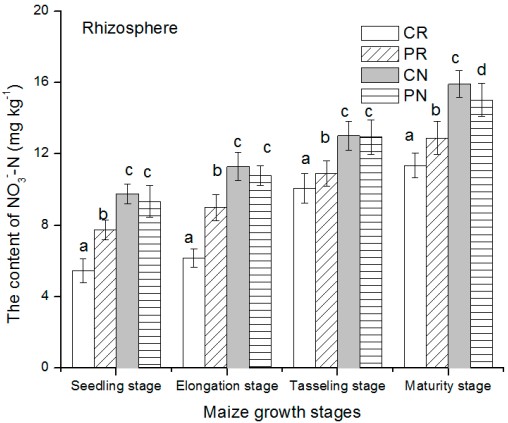 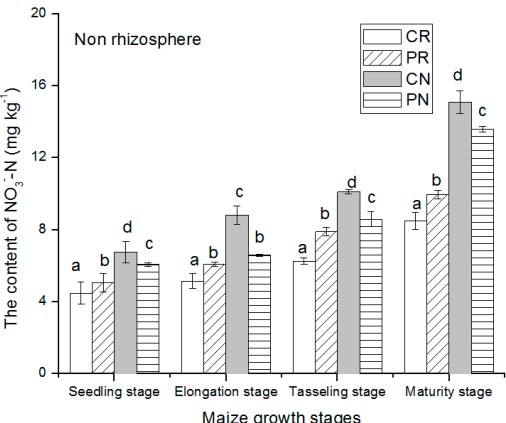

**Figure 2.** $NO_3^-$-N content changes in rhizosphere and non-rhizosphere soil during maize growth stage. Notes: lowercase letters "a, b, c, d" indicate significant differences among different treatments, different lowercase letters above the bars indicate significant differences of total N content.

$NH_4^+$-N contents in the rhizosphere and non-rhizosphere soil with various tillage methods during maize growth are presented above in Figure 3. At first growth stages, $NH_4^+$-N content in the two soil regions was low, then increase gradually with the maize growth until reaching the highest level at physiological maturity period. The content of $NH_4^+$-N in rhizosphere soil and non-rhizosphere soil increased significantly during maize tasseling stage and maturity stage. The rhizosphere soil showed slightly higher concentration of $NH_4^+$-N than non-rhizosphere soil region. In both regions of the soil studied, the CR treatment significantly had the lowest concentration than the other treatments. The content of $NH_4^+$-N during maize growth manifested the same trend CR < PR < PN < CN for rhizosphere and non-rhizosphere soil. Regarding these trends, the CR treatment had 33.9%, 68.4% and, 633.6% lowest than PR, PN and, CN treatments, respectively, under the rhizosphere soil region.

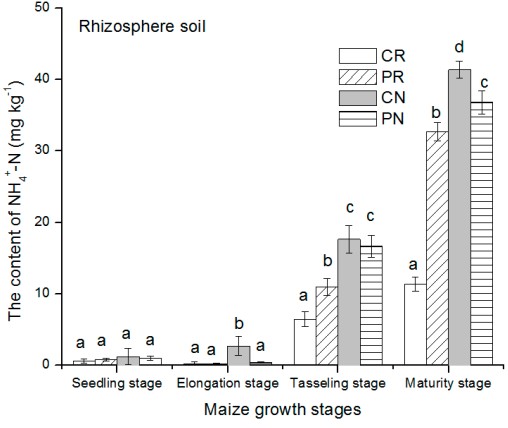 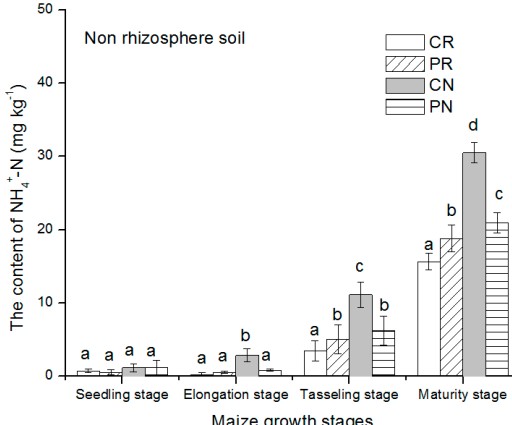

**Figure 3.** $NH_4^+$-N content changes in rhizosphere and non-rhizosphere during maize growth stage. Notes: lowercase letters "a, b, c, d" indicate significant differences among different treatments, different lowercase letters above the bars indicate significant differences of total N content.

### 3.3. Dynamic Change of Urease, Catalase and Alkaline Phosphatase Enzymes Activities in Rhizosphere and Non-Rhizosphere Soil with Various Tillage Methods

The change of the urease enzyme activities in the entire maize growth stage in different treatments under the rhizosphere and non-rhizosphere soil is shown in Figure 4 above. All

the treatments showed the same trend of urease enzyme concentration slightly increased with the age of the crop from the seedling stage to maturity stage in both regions of the soil studied. In comparison to the non-rhizosphere soil region, the rhizosphere soil region exhibited a lower urease amount. This indicated that the increased amount of urease in the non-rhizosphere soil decreased its amount under that rhizosphere soil.

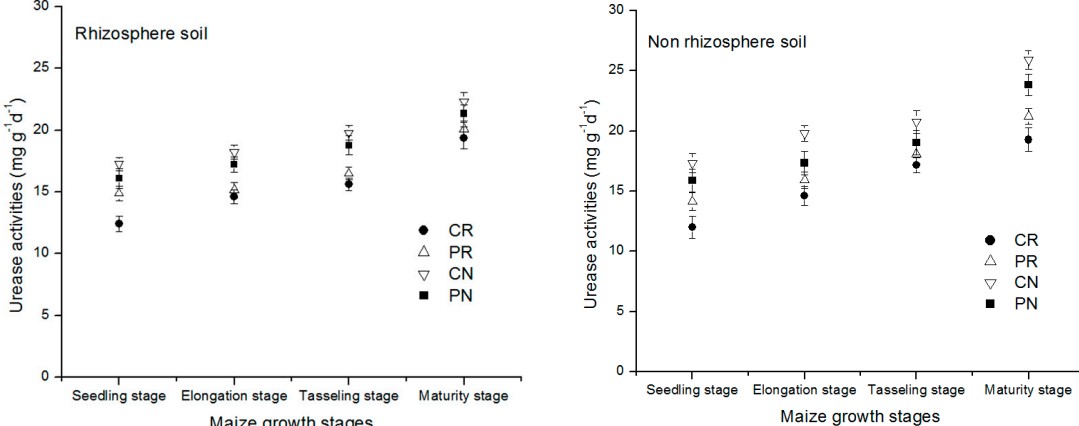

**Figure 4.** Dynamic change of urease enzyme activity in the rhizosphere and non-rhizosphere soil during maize growth stage.

Urease enzymes change in rhizosphere and non-rhizosphere soil during entire maize growth stages was shown in Figure 4. All the treatments showed the slightly increased concentration of urease enzyme with the maize growth. In comparison to non-rhizosphere, a lower urease amount was exhibited in rhizosphere soil. There was vital difference between the treatments for rhizosphere and non-rhizosphere soil. Tendency of the treatments followed the order CN > PN > PR > CR. The CN treatment accounted for the greatest concentration of urease enzyme compared to the other treatments, which were increased by 19.7%, 29.3%, 63.2% in rhizosphere soil and 28.5%, 44.3%, and 66.5% in non-rhizosphere soil at maturity stage of maize.

Dynamic change of catalase enzyme activity in the rhizosphere and non-rhizosphere soil during maize growth stage were shown in Figure 5. A characteristic of the result is the general trend in the catalase enzyme concentration during the growing stages for different treatments. Similarly, the initial catalase amount was higher at the seedling stage then decreased gradually with the age of the maize to the physiological maturity of the maize. The lowest level of catalase enzyme in all treatments becomes visible at the maturity stage. By comparison of non-rhizosphere, the rhizosphere soil had a higher amount of catalase enzyme, meaning that more catalase moved to the rhizosphere soil region of the maize. There was a significant difference between the different treatments. All the treatments found to have the same direction of decreasing with the age of the maize. In all regions of the analyzed, the treatments showed the order CN < PN < PR < CR. At the maturity period, under the rhizosphere soil, the CN treatment had 44.4%, 82.3%, and 109.6% lower than PN, PR, and CR treatments, respectively. Similarly, in the non-rhizosphere soil, the CN treatment had 50.4%, 132.8%, and 170.8% lower than PN, PR, and CR treatments.

Compared with rhizosphere soil, non-rhizosphere soil resulted in increases of alkaline phosphatase by 15.67%, 12.65%, and 26.28% and 8.40% of CR, CN, PR, and PN treatments, respectively. In the whole growing cycle, the alkaline phosphatase amount was the lowest in CN treatment.

There was a significant difference in alkaline phosphatase between the growth stages of growing maize (Figure 6). At the seedling period, a higher concentration of alkaline phosphatase was presented for all treatments, and then were decreased until the maturity period. Compared with rhizosphere soil, non-rhizosphere soil resulted in increases of alkaline phosphatase by 15.67%, 12.65%, and 26.28% and 8.40% of CR, CN, PR, and PN treatments, respectively. The CN treatment showed the lowest amount of alkaline

phosphatase in the whole growing cycle. All these results show that there was an improvement of chemical phosphatase activity in both regions of the studied soil, whereby the CN treatment showed lower alkaline phosphatase compared to others.

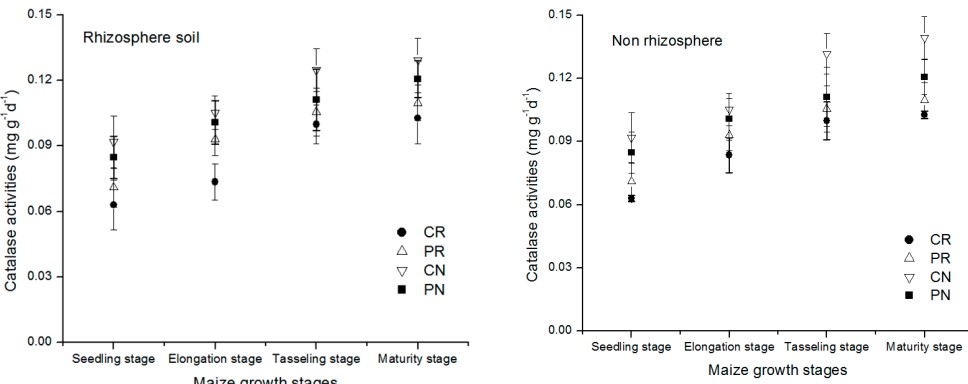

**Figure 5.** Dynamic change of catalase enzyme activity in the rhizosphere and non-rhizosphere soil during maize growth stage.

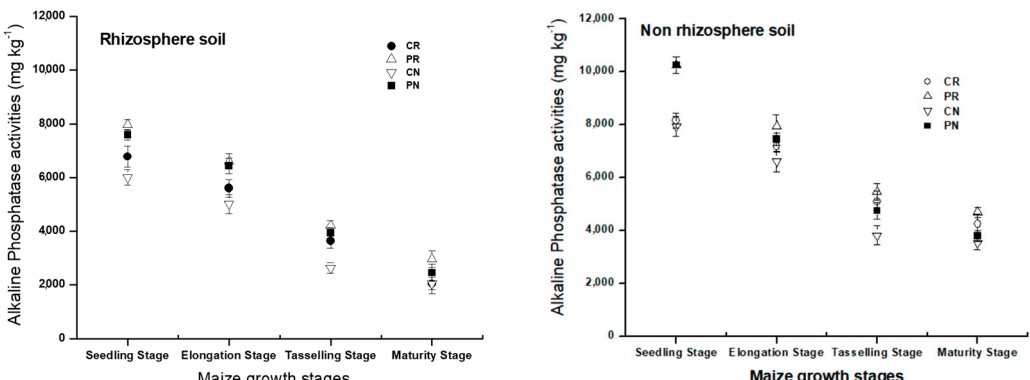

**Figure 6.** Dynamic change of alkaline phosphatase activity in the rhizosphere and non-rhizosphere soil during maize growth stage.

## 4. Discussion

### 4.1. Compare Influence of Different Tillage Methods on N Distribution in Rhizosphere and Non-Rhizosphere Soil

Nitrogen is a major and essential element for crop growth. It's a limiting factor of all crops, and therefore it was an important component in crop production fertilizers [39]. The application of fertilizer in excessive amounts may result in environmental issues like biodiversity loss, global warming, waters eutrophication, and ozone layer depletion. Contrary, less application of nitrogen fertilizer leads to limited yield and, consequently, to low food supply [40,41]. Nitrogen change in soil is greatly dependent on farming system of the tillage method [42]. In this study, the applied tillage methods had an effect on the dynamic change on nitrogen in the soil during the whole maize growth in Figure 1, which increased in the order of CN < PN < PR < CR during all stages of maize growth. Several studies have shown positive effects of non-tillage in combination with rational cropping systems on soil aggregates, organic carbon, nitrogen content, microbial communities, and crop yield [43–45]. A continuous no-tillage approach was a promising practice for increasing nitrogen uptake in maize [46]. Moreover, soil water retention under no tillage condition was beneficial to the crop [47,48]. The application of fertilizer had a vital effect on content of total N in the rhizosphere and non-rhizosphere soils. There was a slight decrease of total N from seedling stage to maturity stage under all soil regions. The rhizosphere soil had slightly higher total N than non-rhizosphere soil, indicating that most of the nutrients shifted to rhizosphere region under all the treatments [49]. Gao et al. also found that the total N remained higher in the rhizosphere compared to the bulk soil [50]

*4.2. Elucidate the Enzyme Activity in Rhizosphere Soil and Non-Rhizosphere Soil with Various Tillage Methods*

The fundamental function of soil enzymes is to catalyze many reactions required for microorganisms to live in the soils, decomposition of organic remains, and the formation of organic matter and soil structure [51]. The extreme cultivation could cause depletion in the microbial biomass and activity [52]. The increase of soil enzyme activities might be the result of soil physical and chemical changes, so there was an immediate relationship between soil enzyme activities and nutrient availability. It has observed that tillage methods significantly influenced the availability and enzyme activities in the rhizosphere and non-rhizosphere soil during the whole maize growth (Figures 4–6). The urease and catalase activities in the rhizosphere soil followed the order CN > PN > PR > CR. No-tillage improved suitable environment for soil enzyme activities [53]. Concerning the effects of tillage, Bendick and Dick (1999) [54] and Mohammadi et al. (2012) [55] reported the decrease of the acid and alkaline phosphomonoesterase, urease and protease activities in the tilled soils, confirming the soil tillage impact No tillage is associated with increased organic matter, improved soil structure and soil microbial activities [56]. This increase was perhaps due to the minimal damage to the soil environment and optimal soil moisture and temperature for microbial growth in zero-till than in plow-till [57,58]. Therefore, no tillage method is an appropriate method to boost the activities of soil enzymes. Enzyme is one of the most active organic components in soil. Soil enzymes and soil microorganism together promote the process of soil metabolism. The enzymes widely present in soil are oxidoreductase and hydrolase, which play an important role in soil fertility. Transformation rate of organic and inorganic nutrients in soil, it mainly depends on the enzymatic activities of invertase, protease, phosphatase, urease, and other hydrolases, as well as oxidoreductase, such as polyphenol oxidase and sulfate reductase.

Under no-tillage practices, soil macroporosity and disturbances were decreased. Reduced or no-tillage practices thus improve soil structure and related biological activities. There are some benefits to reducing soil disturbance such as pear fields. Compared to conventional cultivation methods, they can reduce water losses and prevent soil erosion, and minimize cultivation costs. Reductive soil tillage leads to some modifications in soil physical properties, including moisture content, density, aeration, as well as nutrients or their distribution and the content of organic carbon within soil profile. All these can affect crops to uptake macro- and micronutrients [59]. Many studies have pointed that zero tillage provided many advantages for soil attributes, such as it retains and increase water content in the soil, improve organic matter, cycling of nutrients, reduce soil erosion, enhance micro-organism, soil evaporation reduction, and improved soil fertility [60,61].

## 5. Conclusions

The result revealed that continuous no tillage method is a useful tillage practice to be recommended to the farmers as it ensures the availability of nutrients and improves the soil enzyme activities. Soil enzymes, such as catalase and urease, were beneficial for the improvement of the availability of soil nutrients especially under rhizosphere soil. Rotary tillage and plowing rotary tillage are not recommended practices as it reduced the soil nutrients and the enzyme activity. Studied tillage methods, the effect of continuous no tillage (CN) and plowing no tillage (PN) enabled more nutrients to the rhizosphere soil region; hence, more nutrients were readily available for plants to absorb, and it can be considered as an effective agronomic practice for chemical fertilizer managements.

**Author Contributions:** X.Z. and J.L. (Jinhua Liu) designed the research, N.H. conducted the field and laboratory experiments, J.L. (Jialin Li) and B.S. wrote the paper, X.G. and H.W. reviewed the paper. All authors have read and agreed to the published version of the manuscript.

**Funding:** This research was funded by the Major Projects of Science and Technology of Jilin Province (20200503004SF), Chinese Academy of Engineering Cooperation Project (JL2021-19).

**Data Availability Statement:** Data are contained within the article.

**Conflicts of Interest:** The authors declare no conflict of interest.

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
