# Peer review of "Tillage Methods Change Nitrogen Distribution and Enzyme Activities in Maize Rhizosphere and Non-Rhizosphere Chernozem in Jilin Province of China"

_processes, doi:10.3390/pr11113253_

Round 1
Reviewer 1 Report
Comments and Suggestions for Authors
The objective of this study was to investigate the characteristics of nitrogen and enzyme activities in both rhizosphere and non-rhizosphere soil with four different tillage methods in maize production in China. The study has scientific merit; however, there are several issues that need to be addressed.
I don’t see the line numbers on the manuscript. So, it was a little tricky to provide a specific comment on the manuscript. Please add the line numbers so that it will be earlier to provide the comments.
The English language should be improved as well.
1. Introduction
Paragraph 1, lines 5-6: “The non-rhizosphere region, sometimes called bulky soil, is the soil free zone of plant roots, microbes and which is not part of any rhizosphere region”. I think this sentence is not correct. The zone is not soil free. The correct definition should be: The non-rhizosphere region, sometimes called bulky soil, is the soil layer free of plant roots and microbes and not part of any rhizosphere region.
Paragraph 2, lines 1-2: “Land productivity and soil fertility greatly depend on continued utilization of inorganic fertilizers.” I don’t agree with this statement. Inorganic fertilizers are not used in organic farming, and organic fertilizers like manure are an alternative to inorganic fertilizers. Land productivity and soil fertility depend on factors like soil quality, management practices, and the type of crops.
2. Materials and Methods
What is plowing-no tillage? If you plow, how is it no-tillage? My understanding is that plowing is a tillage practice. Please provide a short overview and the specific field operation carried out under each tillage practice (CR, CN, PR, and PN).
4. Discussion
This section is very brief. It seems like the authors summarized the results of this study in this section. I was hoping they would compare their findings with other similar studies. Please also provide some insights on what still unknown and what future studies should investigate.
Comments on the Quality of English LanguageI noted several grammatical errors and awkward sentences in the manuscript. Hence, the English language should be improved.
Author Response
Reviewer 1
Comments and Suggestions for Authors
The objective of this study was to investigate the characteristics of nitrogen and enzyme activities in both rhizosphere and non-rhizosphere soil with four different tillage methods in maize production in China. The study has scientific merit; however, there are several issues that need to be addressed.
I don’t see the line numbers on the manuscript. So, it was a little tricky to provide a specific comment on the manuscript. Please add the line numbers so that it will be earlier to provide the comments.
the line numbers have been added on the manuscript
The English language should be improved as well.
- Introduction
Paragraph 1, lines 5-6: “The non-rhizosphere region, sometimes called bulky soil, is the soil free zone of plant roots, microbes and which is not part of any rhizosphere region”. I think this sentence is not correct. The zone is not soil free. The correct definition should be: The non-rhizosphere region, sometimes called bulky soil, is the soil layer free of plant roots and microbes and not part of any rhizosphere region.
The sentence has been corrected to be “The non-rhizosphere region, sometimes called bulky soil, is the soil layer free of plant roots and microbes and not part of any rhizosphere region.”
Paragraph 2, lines 1-2: “Land productivity and soil fertility greatly depend on continued utilization of inorganic fertilizers.” I don’t agree with this statement. Inorganic fertilizers are not used in organic farming, and organic fertilizers like manure are an alternative to inorganic fertilizers. Land productivity and soil fertility depend on factors like soil quality, management practices, and the type of crops.
The sentence has been corrected to be “Land productivity and soil fertility depend on factors like soil quality, fertilizers, management practices, and the type of crops.”
- Materials and Methods
What is plowing-no tillage? If you plow, how is it no-tillage? My understanding is that plowing is a tillage practice. Please provide a short overview and the specific field operation carried out under each tillage practice (CR, CN, PR, and PN).
CR, CN, PR, and PN have been described in the text.
CR: rotary tillage every year, the soil tilling depth was approximately 10-12 cm, no-tillage seeder sowing, and fertilizer; CN: seeder sowing and fertilizer at approximately 20 cm soil depth, direct use of a no-tillage seeder for sowing and fertilization without other treatments; PR: plowing at a depth of approximately 20 cm in the first year and the same treatment as CR in the last two years; PN: plowing and tillage at approximately 20 cm depth in the first year and the same treatment as CN in the second year.
- Discussion
This section is very brief. It seems like the authors summarized the results of this study in this section. I was hoping they would compare their findings with other similar studies. Please also provide some insights on what still unknown and what future studies should investigate.
We have compared their findings with other similar studies, and references 40,41,42, 47, 51, 52, 54, 55 have been added in the text.
Comments on the Quality of English Language
I noted several grammatical errors and awkward sentences in the manuscript. Hence, the English language should be improved.
We have check the paper.
Reviewer 2 Report
Comments and Suggestions for Authors
The work Processes-2619751 submitted for review is of certain scientific interest.
1.However, the authors did not quite methodically clear present "Materials and Methods" of the experiment. Please, present Chapter 2.2. and 2.3 in more detail.
2. Results and discussion.
The statistical analysis of the data is not well presented and discussed clearly. Detailed information is required.
No information available on the change in total nitrogen and enzyme activity by study year (2019-2021).
3. The Discussion section needs to be presented in more detail. It is necessary to provide data from other researchers on the problem under study
4. The whole paper should be proofread to check minor sentence structure.
Author Response
Reviewer 2
Comments and Suggestions for Authors
The work Processes-2619751 submitted for review is of certain scientific interest.
- However, the authors did not quite methodically clear present "Materials and Methods" of the experiment. Please, present Chapter 2.2. and 2.3 in more detail.
Chapter 2.2. and 2.3 have added more detail.
- Results and discussion.
The statistical analysis of the data is not well presented and discussed clearly. Detailed information is required.No information available on the change in total nitrogen and enzyme activity by study year (2019-2021).
The experiment of different tillage treatments was carried out for 3 years (2019-2021). Soil samples were collected in 2021october, in order to investigate the difference of nitrogen content and enzyme activity between different tillage treatments.
3.The Discussion section needs to be presented in more detail. It is necessary to provide data from other researchers on the problem under study
We have compared their findings with other similar studies, and references 40,41,42, 47, 51, 52, 54, 55 have been added in the text.
- The whole paper should be proofread to check minor sentence structure We have check the manuscript.
Reviewer 3 Report
Comments and Suggestions for Authors
"The abstract's structure is inadequate, as it is too general and fails to effectively summarize the paper's discussions. The introduction section provides only very general information and should be expanded. Additionally, there are relevant reports in the existing literature that should be referenced in the paper.
One notable absence is the lack of a research hypothesis in the work. It is necessary to provide a more detailed description of the area where the experiment was conducted. A map indicating the region would be useful.
What is the average consumption of inorganic fertilizers per hectare in China? Which variety of corn was used in the experiment?
Furthermore, the statistical analysis needs to be supplemented, and the figures are not very legible.
Why does the paper have a "Results and discussion" section followed by a separate "Discussion" section? Such divisions may confuse the reader. There is no discussion in the "Results and discussion" section.
The presentation of results and their interpretation is rather weak. The discussion is primarily focused on presenting observed results and lacks a substantial effort to explain the underlying causes of the observed patterns. There is practically no discussion. It is necessary to address why the applied tillage methods had an effect on the dynamic change in nitrogen.
The literature cited in the article should be more recent.
The summary requires rephrasing."
Author Response
Reviewer 3
Comments and Suggestions for Authors
- "The abstract's structure is inadequate, as it is too general and fails to effectively summarize the paper's discussions. The introduction section provides only very general information and should be expanded. Additionally, there are relevant reports in the existing literature that should be referenced in the paper.
The abstract has been revised according to the comment.
- One notable absence is the lack of a research hypothesis in the work. It is necessary to provide a more detailed description of the area where the experiment was conducted. A map indicating the region would be useful.
research hypothesis in the work” Farming practices can alter the physical and chemistry environment of the soil and inevitably affect the distribution of nitrogen and enzymes in the rhizosphere” has been added.
- What is the average consumption of inorganic fertilizers per hectare in China? Which variety of corn was used in the experiment?
800 kg·ha-1 combined fertilizer (N: P2O5:K2O=26%:11%:11%) was applied.
- Furthermore, the statistical analysis needs to be supplemented, and the figures are not very legible.
- Why does the paper have a "Results and discussion" section followed by a separate "Discussion" section? Such divisions may confuse the reader. There is no discussion in the "Results and discussion" section.
"Results and discussion" section has been changed to be “Results”
- The presentation of results and their interpretation is rather weak. The discussion is primarily focused on presenting observed results and lacks a substantial effort to explain the underlying causes of the observed patterns. There is practically no discussion. It is necessary to address why the applied tillage methods had an effect on the dynamic change in nitrogen.
- The literature cited in the article should be more recent.
We have compared their findings with other similar studies, and references 40,41,42, 47, 51, 52, 54, 55 have been added in the text.
- The summary requires rephrasing."
The abstract has been revised according to the comment.
Reviewer 4 Report
Comments and Suggestions for Authors
General comment:
This field study addresses an important aspect of farming systems – the impact of different tillage methods on nitrogen content and enzyme activities in rhizosphere soil. It highlights the potential benefits of certain tillage practices for nutrient availability and soil health.
Specific comments:
-
The text effectively conveys its objective, which is to investigate the characteristics of nitrogen and enzyme activities in rhizosphere soil under different tillage methods. This clarity helps readers understand the study's purpose.
-
The text mentions the four treatment plots established for the study, which is crucial information. However, providing more details about the methodology, such as the dataset, number of plots or replicates per treatment, would enhance the reader's understanding of the study design.
-
While the text mentions that soil enzymes were more active in rhizosphere soil, it would be helpful to discuss the implications of this finding. How might increased enzyme activity benefit plant growth and soil health?
- While the text suggests that continuous no-till (CN) is recommended for improving nutrient availability and soil enzyme activities, it would be beneficial to provide practical recommendations for farmers based on the study's results.
Constructive feedback:
As mentioned earlier, providing more details about the study's methodology, including the sampling process, measurement techniques, and statistical analysis, would strengthen the study's rigor and allow readers to assess the validity of the results. The text touches upon avoiding environmental pollution through tillage methods, but it would be valuable to expand on this aspect. How do the findings relate to environmental sustainability and pollution prevention?
Summary:
In summary, this study investigates the impact of different tillage methods on nitrogen content and enzyme activities in rhizosphere soil. It effectively communicates its objective and presents data on these parameters. To enhance the text, consider providing more methodological details, discussing environmental implications, and offering practical recommendations for farmers based on the findings.
Author Response
Reviewer 4
General comment:
This field study addresses an important aspect of farming systems – the impact of different tillage methods on nitrogen content and enzyme activities in rhizosphere soil. It highlights the potential benefits of certain tillage practices for nutrient availability and soil health.
Specific comments:
- The text effectively conveys its objective, which is to investigate the characteristics of nitrogen and enzyme activities in rhizosphere soil under different tillage methods. This clarity helps readers understand the study's purpose.
- The text mentions the four treatment plots established for the study, which is crucial information. However, providing more details about the methodology, such as the dataset, number of plots or replicates per treatment, would enhance the reader's understanding of the study design.
“There were three replicates per treatment in 12 subplots of 500 m2”
- While the text mentions that soil enzymes were more active in rhizosphere soil, it would be helpful to discuss the implications of this finding. How might increased enzyme activity benefit plant growth and soil health?
“Enzyme is one of the most active organic components in soil. Soil enzyme and soil microorganism together promote the process of soil metabolism. The enzymes widely present in soil are oxidoreductase and hydrolase, which play an important role in soil fertility. Transformation rate of organic and inorganic nutrients in soil, it mainly depends on the enzymatic activities of invertase, protease, phosphatase, urease and other hydrolases as well as oxidoreductase such as polyphenol oxidase and sulfate reductase.” has been added in the text.
- While the text suggests that continuous no-till (CN) is recommended for improving nutrient availability and soil enzyme activities, it would be beneficial to provide practical recommendations for farmers based on the study's results.
Constructive feedback:
As mentioned earlier, providing more details about the study's methodology, including the sampling process, measurement techniques, and statistical analysis, would strengthen the study's rigor and allow readers to assess the validity of the results. The text touches upon avoiding environmental pollution through tillage methods, but it would be valuable to expand on this aspect. How do the findings relate to environmental sustainability and pollution prevention?
The research objectives and significance was rewritten “This work seeks to find distribution of nitrogen and enzymatic activities in rhizosphere and non-rhizosphere soil of maize with diverse tillages. The results will provide a theoretical basis for the establishment of rational straw returning and tillage methods in the study area.”
Round 2
Reviewer 2 Report
Comments and Suggestions for Authors
Thanks to the authors for answering my questions. In general, the authors made edits according to the my comments. However, in the "Results" section, the authors did not discuss the statistical analysis of the data in the text. I suggest the authors improve this section of the manuscript.
Author Response
Reviewer 2
Comments and Suggestions for Authors
Thanks to the authors for answering my questions. In general, the authors made edits according to the my comments. However, in the "Results" section, the authors did not discuss the statistical analysis of the data in the text. I suggest the authors improve this section of the manuscript.
Re: Statistical analysis on the data has been done and significant differences was marked the in Figure1, Figure2, Figure3.
Reviewer 3 Report
Comments and Suggestions for Authors
Accept
Author Response
Thanks for your comment.
Reviewer 4 Report
Comments and Suggestions for Authors
The research highlights the importance of tillage choices in optimizing nutrient availability for crops and reducing environmental pollution. However, the study has limitations, including a narrow scope, a short duration, and incomplete data for one treatment group (PN). To enhance the study's credibility, addressing these limitations and considering external factors in future research is recommended.
Author Response
Reviewer 4
Comments and Suggestions for Authors
The research highlights the importance of tillage choices in optimizing nutrient availability for crops and reducing environmental pollution. However, the study has limitations, including a narrow scope, a short duration, and incomplete data for one treatment group (PN). To enhance the study's credibility, addressing these limitations and considering external factors in future research is recommended.
Re: Thanks for the valuable comments. In future research, more external factors will be considered, especially to enrich the data about PN.